# CURVE YOUR ATTENTION:
# MIXED-CURVATURE TRANSFORMERS
# FOR GRAPH REPRESENTATION LEARNING

## ABSTRACT

Real-world graphs naturally exhibit hierarchical trees and cyclic structures that are unfit for the typical Euclidean space. While there exist graph neural networks that utilize hyperbolic or spherical spaces towards embedding such structures more accurately, these methods are confined under the message-passing paradigm, making them vulnerable against side-effects such as oversmoothing and oversquashing. More recent work have proposed global attention-based graph Transformers that can alleviate such drawbacks and easily model long-range interactions, but their extensions towards non-Euclidean geometry are yet unexplored. To bridge this gap, we propose Fully Product-Stereographic Transformer, a generalization of Transformers towards operating entirely on the product of constant curvature spaces. When combined with tokenized graph Transformers, our model can learn the curvature appropriate for the input graph in an end-to-end fashion, without any additional tuning on different curvature initializations. We also provide a kernelized approach to non-Euclidean attention, which enables our model to run with computational cost linear to the number of nodes and edges while respecting the underlying geometry. Experiments on graph reconstruction and node classification demonstrate the benefits of generalizing Transformers to the non-Euclidean domain.

## 1 INTRODUCTION

Learning from graph-structured data is a challenging task in machine learning, with various downstream applications that involve modeling individual entities and relational interactions among them (Sen et al., 2008; Watts & Strogatz, 1998; Gleich et al., 2004). A dominant line of work consists of graph convolutional networks (GCNs) that aggregate features across graph neighbors through *message-passing* (Gilmer et al., 2017; Kipf & Welling, 2016; Veličković et al., 2017; Wu et al., 2019; Hamilton et al., 2017). While most GCNs learn features that lie on the typical Euclidean space with zero curvature, real-world graphs often comprise of complex structures such as hierarchical trees and cycles that Euclidean space requires excessive dimensions to accurately embed (Sala et al., 2018). In response, the graph learning community has developed generalizations of GCNs to spaces with non-zero curvature such as hyperbolic, spherical, or mixed-curvature spaces with both negative and positive curvatures (Chami et al., 2019; Liu et al., 2019; Bachmann et al., 2020; Xiong et al., 2022).

Unfortunately, non-Euclidean GCNs are not immune to harmful side-effects of message-passing such as oversmoothing (Oono & Suzuki, 2019; Cai & Wang, 2020; Yang et al., 2022) and oversquashing (Topping et al., 2021; Alon & Yahav, 2020). These drawbacks make it difficult to stack GCN layers towards large depths, limiting its expressive power (Feng et al., 2022; Maron et al., 2019) as well as predictive performance on tasks that require long-range interactions to solve (Dwivedi et al., 2022; Liu et al., 2021). To cope with such limitations, recent work have instead proposed Transformer-based graph encoders that can easily exchange information across long-range distances through global self-attention (Kim et al., 2022; Ying et al., 2021; Dwivedi & Bresson, 2020; Kreuzer et al., 2021). However, existing graph Transformers are still confined within the Euclidean regime, and their extensions towards non-Euclidean geometry has not yet been studied.

In this paper, we bridge this gap by generalizing the Transformer architecture (Vaswani et al., 2017) towards non-Euclidean spaces with learnable curvatures. Specifically, we endow each attention head a stereographic model (Bachmann et al., 2020) that can universally represent Euclidean, hyperbolic, and

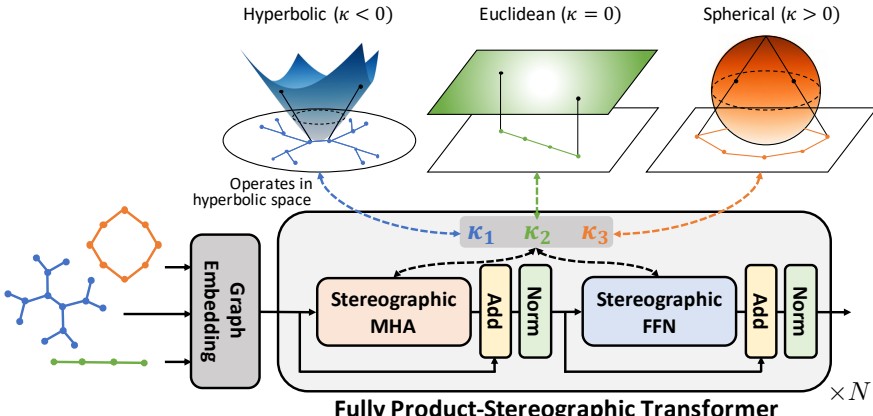

Figure 1: Illustration of our proposed FPS-T architecture. Well-known constant curvature spaces can be projected to the stereographic model, with a common chart map isomorphic to the $d$-dimensional Euclidean space. Each space can efficiently embed different types of graphs (*e.g.*, trees in hyperbolic space, lines in Euclidean space, and cycles in spherical space). In FPS-T, each layer chooses a set of curvatures that fits the input graph by changing the sign of the curvature $\kappa$ in a differentiable manner.

spherical spaces (Figure 1). We generalize each operation of the Transformer architecture to inputs on the product-stereographic model, all of which are end-to-end differentiable with respect to the curvatures, thereby allowing the model to jointly train embeddings as well as the underlying curvature. The resulting model, which we name as **Fully Product-Stereographic Transformer (FPS-T)**, takes advantage of both non-Euclidean geometry and long-range interactions. We empirically show that the learnable sectional curvature of FPS-T successfully converges to the geometry of the input graph, leading to better predictive performance and parameter efficiency in graph reconstruction and node classification compared to its Euclidean counterpart. To the best of our knowledge, our work is the first to propose a natural generalization of Transformers to mixed-curvature spaces. We summarize our core contributions as follows:

- We propose FPS-T, a generalization of Transformer towards operating entirely on the product-stereographic model with curvatures that are learnable in an end-to-end fashion.

- For graph representation learning, we integrate FPS-T with Tokenized Graph Transformer (Kim et al., 2022), and develop a kernelized approximation of non-Euclidean attention to reduce the computational cost to linear in number of nodes and edges.

- Graph reconstruction and node classification experiments on synthetic as well as real-world graphs demonstrate the benefit of generalizing Transformers to the mixed-curvature domain.

## 2 RELATED WORK

**Non-Euclidean graph representations.** Non-Euclidean spaces are known to well-preserve specific types of graph structure where Euclidean space fails. Especially, non-Euclidean spaces with constant sectional curvature, *e.g.*, hyperbolic and spherical spaces, are widely used in graph representation learning due to its tractable operations. Hyperbolic spaces are capable of efficiently embedding complex hierarchical structures in graphs (Nickel & Kiela, 2018; 2017; Ganea et al., 2018; Krioukov et al., 2010; Sala et al., 2018). Graphs with cyclic structures are well-suited for spherical spaces (Wilson et al., 2014; Grattarola et al., 2019). Riemannian manifolds with varying curvature and constant sign are also proposed for graph encoding (Cruceru et al., 2021). However, Riemannian manifolds where the sign of the curvature is fixed are not a good choice for more complex graphs that exhibit both hierarchy and cycles. Instead, the product of constant-curvature spaces (Gu et al., 2019), heterogeneous manifolds (Giovanni et al., 2022), and pseudo-Riemannian manifolds (Law & Stam, 2020) are found to be well-suited for learning representations of such complex graphs.

Message passing GCNs also benefit from considering a non-Euclidean representation space. Hyperbolic GCNs are known to outperform Euclidean counterparts in various tasks on hierarchical graphs such as citation networks (Chami et al., 2019; Zhang et al., 2021; Pei et al., 2020) and

molecules (Chami et al., 2019; Liu et al., 2019). Deepsphere (Defferrard et al., 2020) also adopted the spherical space to GCNs with applications such as 3D object and earth climate modeling. To take the advantage of multiple spaces, (Zhu et al., 2020b) proposed a hybrid architecture that fuses Euclidean and hyperbolic graph representations together. (Deng et al., 2023) similarly proposed modeling interactions between three constant-curvature spaces (*i.e.*, Euclidean, hyperbolic, and spherical). To allow smooth connections between the three constant-curvature spaces, (Bachmann et al., 2020) proposed a model of constant-curvature space called the stereographic model, on which geometric operations such as distances and inner products are differentiable at all curvature values including zero. Incorporating pseudo-Riemannian manifolds with the GCN architecture also showed promising results (Xiong et al., 2022), but its performance is sensitive to the time dimension of the manifold, which requires extensive hyperparameter tuning.

Overall, GCNs achieve great predictive performance in homophilic graphs where connected nodes share the same features, but they tend to fail in hetereophilic graphs, as stacking up GCN layers to capture message passing between distant nodes induces oversmoothing (Oono & Suzuki, 2019; Cai & Wang, 2020) and oversquashing (Topping et al., 2021). To relieve this architectural limitation while utilizing non-Euclidean geometrical priors, we instead develop a Transformer-based graph encoder that operates on the steregraphic model to learn graph representations.

**Graph Transformers.** Inspired by huge success of Transformers in NLP and CV (Devlin et al., 2018; Brown et al., 2020; Dosovitskiy et al., 2020), there exist various work that extend Transformers for encoding graphs with edge connectivities that are neither sequential nor grid-like. Graph Transformer (Dwivedi & Bresson, 2020) and Spectral Attention Network (Kreuzer et al., 2021) were the first pioneers to explore this direction by replacing sinusoidal positional encodings widely used in NLP with Laplacian eigenvectors of the input graph. Graphormer (Ying et al., 2021) then proposed utilizing edge connectivities by using shortest-path distances as an attention-bias, showing state-of-the-art performance on molecular property prediction. TokenGT (Kim et al., 2022) proposed a tokenization technique that views each graph as a sequence of nodes and edges. Unlike other methods, TokenGT allows straightforward integration of engineering techniques of pure Transformers such as linearized attention (Katharopoulos et al., 2020), while enjoying theoretical expressivity that surpasses that of message-passing GCNs.

However, existing graph Transformer architectures are yet confined within the Euclidean domain, making them unable to precisely embed graphs onto the feature space similarly to geometric GCNs. While Hyperbolic Attention Network (Gulcehre et al., 2018) proposed an attention mechanism that operates on hyperbolic space, its distance-based approach imposes a computational cost quadratic to the graph size and its geometry is fixed to hyperbolic. Instead, we generalize the representation space of Transformer to stereographic model, which allows us to cover more various types of graphs. We also linearize the attention mechanism on the stereographic model similar to Katharopoulos et al. (2020), which results in a model that runs in cost linear to the number of nodes and edges.

## 3 PRELIMINARIES

In this section, we introduce concepts related to our main geometrical tool, the product-stereographic model (Bachmann et al., 2020). We also discuss multi-head attention, the driving force of the Transformer architecture (Vaswani et al., 2017).

### 3.1 PRODUCT-STEREOGRAPHIC MODEL

**Riemannian manifolds.** A Riemannian manifold is consisted of a smooth manifold $\mathcal{M}$ and a metric tensor $g$. Each point $\boldsymbol{x}$ on the manifold $\mathcal{M}$ defines a tangent space $\mathcal{T}_{\boldsymbol{x}}\mathcal{M}$, which is a collection of all vectors that are tangent to $\boldsymbol{x}$, also called the tangent vector. The metric tensor $g : \mathcal{M} \to \mathbb{R}^{n \times n}$ assigns a positive-definite matrix to each point $\boldsymbol{x}$, which defines its inner product $\langle \cdot, \cdot \rangle_{\boldsymbol{x}} : \mathcal{T}_{\boldsymbol{x}}\mathcal{M} \times \mathcal{T}_{\boldsymbol{x}}\mathcal{M} \to \mathbb{R}$ as $\boldsymbol{v}_1^T g(\boldsymbol{x}) \boldsymbol{v}_2$ where $\boldsymbol{v}_1, \boldsymbol{v}_2 \in \mathcal{T}_{\boldsymbol{x}}\mathcal{M}$ are the tangent vectors of $\boldsymbol{x}$. The metric tensor also defines geometrical properties and operations on the Riemannian manifold. Geodesic $\gamma$ is the shortest curve between two points $\boldsymbol{x}, \boldsymbol{y} \in \mathcal{M}$ and its distance can be computed as $d_{\mathcal{M}}(\boldsymbol{x}, \boldsymbol{y}) = \int_0^1 \langle \dot{\gamma}(t), \dot{\gamma}(t) \rangle_{\gamma(t)} dt$, where $\gamma : [0, 1] \to \mathcal{M}$ is a unit-speed curve satisfying $\gamma(0) = \boldsymbol{x}$ and $\gamma(1) = \boldsymbol{y}$. We can move the point $\boldsymbol{x} \in \mathcal{M}$ along a tangent vector $\boldsymbol{v} \in \mathcal{T}_{\boldsymbol{x}}\mathcal{M}$ using exponential map $\exp_{\boldsymbol{x}} : \mathcal{T}_{\boldsymbol{x}}\mathcal{M} \to \mathcal{M}$ which is defined as $\exp_{\boldsymbol{x}}(\boldsymbol{v}) = \gamma(1)$ where $\gamma$ is a geodesic and $\gamma(0) = \boldsymbol{x}, \gamma(0) = \boldsymbol{v}$. The logarithmic map $\log_{\boldsymbol{x}} : \mathcal{M} \to \mathcal{T}_{\boldsymbol{x}}\mathcal{M}$ is the inverse of $\exp_{\boldsymbol{x}}$. A tangent vector $\boldsymbol{v} \in \mathcal{T}_{\boldsymbol{x}}\mathcal{M}$ can be transferred along a geodesic from $\boldsymbol{x}$ to $\boldsymbol{y}$ using parallel transport $\mathrm{PT}_{\boldsymbol{x} \to \boldsymbol{y}} : \mathcal{T}_{\boldsymbol{x}}\mathcal{M} \to \mathcal{T}_{\boldsymbol{y}}\mathcal{M}$.

Note that the product of Riemannian manifolds is also a Riemannian manifold. A point on the product Riemannian manifold $\boldsymbol{x} \in \otimes_{i=1}^n \mathcal{M}_i$ is consisted of points from each component manifold $\mathcal{M}_i$ as $\boldsymbol{x} = \|_{i=1}^n \boldsymbol{x}_i$, where $\boldsymbol{x}_i \in \mathcal{M}_i$ and $\|$ denotes concatenation. The distance between $\boldsymbol{x}, \boldsymbol{y} \in \otimes_{i=1}^n \mathcal{M}_i$ is calculated as $\sqrt{\sum_{i=1}^n d_{\mathcal{M}_i}^2(\boldsymbol{x}_i, \boldsymbol{y}_i)}$. Exponential/logarithmic maps and parallel transports are applied in a manifold-wise fashion (*e.g.*, $\exp_{\boldsymbol{x}}(\boldsymbol{v}) = \|_{i=1}^n \exp_{\boldsymbol{x}_i}(\boldsymbol{v}_i)$ with $\boldsymbol{v} = \|_{i=1}^n \boldsymbol{v}_i$ and $\boldsymbol{v}_i \in \mathcal{T}_{\boldsymbol{x}_i} \mathcal{M}_i$).

**Constant-curvature spaces.** Curvature is an important geometrical property used to characterize Riemannian manifolds. One widely-used curvature to explain Riemannian manifolds is the sectional curvature: given two linearly independent tangent vector fields $U, V \in \mathfrak{X}(\mathcal{M})$, the sectional curvature $K(U, V)$ is computed as $K(U, V) = \frac{\langle R(U,V)V,U \rangle}{\langle U,U \rangle \langle V,V \rangle - \langle U,V \rangle^2}$, where $R(\cdot, \cdot) : \mathfrak{X}(\mathcal{M}) \times \mathfrak{X}(\mathcal{M}) \times \mathfrak{X}(\mathcal{M}) \to \mathfrak{X}(\mathcal{M})$ is a Riemannian curvature tensor. The sectional curvature measures the divergence between geodesics starting with the tangent vector fields $U, V$ for each point of the manifold. With positive or negative sectional curvatures, geodesics become closer or farther than with zero curvature. Throughout this paper, we refer to a space of a constant sectional curvature as a *constant-curvature space*. For example, the Euclidean space $\mathbb{E}$ is the special case of the constant-curvature space with zero curvature. When positive or negative, we call the corresponding spaces to be hyperbolic $\mathbb{H}$ or spherical $\mathbb{S}$.

**Stereographic models.** A $d$-dimensional stereographic model $\mathfrak{st}_\kappa^d$ is a constant-curvature space with curvature $\kappa \in \mathbb{R}$. One attractive property of the stereographic model is that the operations such as distance, exp/log-map, and parallel transport are differentiable at any curvature value $\kappa$, including $\kappa = 0$. This enables the stereographic model to learn the curvature value $\kappa$ without any constraint.

The manifold of the stereographic model $\mathfrak{st}_\kappa^d$ is $\{\boldsymbol{x} \in \mathbb{R}^d | -\kappa \|\boldsymbol{x}\|^2 < 1\}$. The metric tensor is defined as $g^\kappa(\boldsymbol{x}) = \frac{4}{1+\kappa\|\boldsymbol{x}\|^2} \boldsymbol{I} =: (\lambda_{\boldsymbol{x}}^\kappa)^2 \boldsymbol{I}$, where $\lambda_{\boldsymbol{x}}^\kappa$ is known as the conformal factor. The mobius addition between two points $\boldsymbol{x}, \boldsymbol{y} \in \mathfrak{st}_\kappa^d$ is computed as $\boldsymbol{x} \oplus_\kappa \boldsymbol{y} = \frac{(1-2\kappa\boldsymbol{x}^T\boldsymbol{y}-\kappa\|\boldsymbol{y}\|^2)\boldsymbol{x}+(1+\kappa\|\boldsymbol{x}\|^2)\boldsymbol{y}}{1-2\kappa\boldsymbol{x}^T\boldsymbol{y}+\kappa^2\|\boldsymbol{x}\|^2\|\boldsymbol{y}\|^2}$. Based on mobius addition, we can derive other geometric operations as Table 3 in Appendix A. The table also shows that when $\kappa$ converges to zero, the operations become equivalent to Euclidean space operations, so the stereographic model essentially recovers Euclidean geometry.

## 3.2 Multi-Head Attention

In vanilla Transformer (Vaswani et al., 2017), each block consists of multiple attention heads, each taking a sequence of token embeddings $\boldsymbol{X} \in \mathbb{R}^{n \times d}$ with sequence length $n$ and feature dimension $d$ as input. Three linear layers $\boldsymbol{W}^Q, \boldsymbol{W}^K, \boldsymbol{V}^V \in \mathbb{R}^{d \times d'}$ first map each token embedding into queries $\boldsymbol{Q}$, keys $\boldsymbol{K}$, and values $\boldsymbol{V}$ with head-dimension $d'$, respectively. Then, the attention score matrix is computed by scaled Euclidean dot-product between $\boldsymbol{Q}$ and $\boldsymbol{K}$, followed by row-wise softmax activation $\sigma(\cdot)$. The attention score matrix is then multiplied to value $\boldsymbol{V}$, returning contextualized token embeddings. The overall procedure can be written as

$$\boldsymbol{Q} = \boldsymbol{X}\boldsymbol{W}^Q, \quad \boldsymbol{K} = \boldsymbol{X}\boldsymbol{W}^K, \quad \boldsymbol{V} = \boldsymbol{X}\boldsymbol{W}^V, \quad \text{Attn}(\boldsymbol{X}) = \sigma\left(\frac{\boldsymbol{Q}\boldsymbol{K}^T}{\sqrt{d'}}\right)\boldsymbol{V}. \tag{1}$$

The output from multiple attention heads are concatenated together, then processed through a feed-forward layer before proceeding to the next Transformer block.

## 4 Fully Product-Stereographic Transformer

Here, we describe the inner wirings of our proposed method. We generalize each operation in Transformer to the product-stereographic model, together forming a geometric Transformer architecture that operates entirely within the stereographic model.

### 4.1 Stereographic Neural Networks

We first introduce the stereographic analogies of the Euclidean neural networks such as the linear layer, activation, layer normalization, and logit functions. We denote the product-stereographic model $\otimes_{i=1}^H \mathfrak{st}_{\kappa_i}^d$ as $\mathfrak{st}_{\otimes\boldsymbol{\kappa}}^d$, where $\boldsymbol{\kappa} = (\kappa_1, \ldots, \kappa_H)$ is the ordered set of curvatures of $d$-dimensional component spaces within a Transformer block with $H$ attention heads. We also use the superscript $\otimes\boldsymbol{\kappa}$ to denote Riemannian operations on product-stereographic model that decompose representations into equal parts, apply the operation, then concatenate back to the product space (*e.g.*, if $\boldsymbol{v} = [v_1, \ldots, v_H]$, then $\exp_{\boldsymbol{0}}^{\otimes\boldsymbol{\kappa}}(\boldsymbol{v}) := \|_{i=1}^H \exp_{\boldsymbol{0}}^{\kappa_i}(v_i)$).

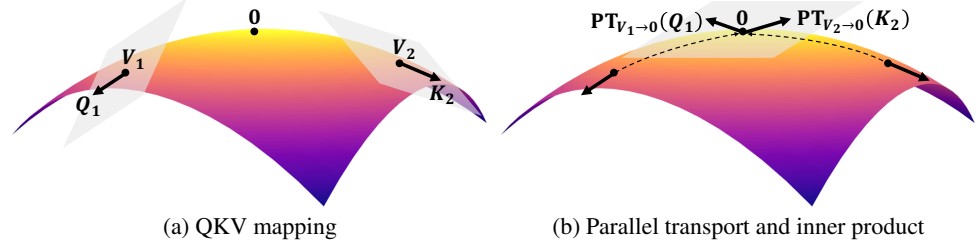

(a) QKV mapping       (b) Parallel transport and inner product

Figure 2: Illustration of our attention mechanism on the non-Euclidean space. FPS-T considers each value-vector as a point that resides on the stereographic model, and query/key-vectors as tangent vectors on the corresponding tangent spaces. All query/key-vectors are parallel-transported to the origin prior to dot-product attention, thereby taking the given geometry into account.

**Stereographic linear layer, activation, and layer normalization.** Given a Euclidean neural network $f$, we can define its stereographic counterpart as $\exp_{\mathbf{0}}^{\otimes\boldsymbol{\kappa}}\left(f\left(\log_{\mathbf{0}}^{\otimes\boldsymbol{\kappa}}(\boldsymbol{X})\right)\right)$. The stereographic linear layer $\mathrm{Linear}_{\otimes\boldsymbol{\kappa}}(\boldsymbol{X};\boldsymbol{W})$ is thus defined by setting $f$ as the Euclidean linear layer $f(\boldsymbol{X};\boldsymbol{W}) = \boldsymbol{X}\boldsymbol{W}$. The same approach can be used for any Euclidean activation function $f_{\mathrm{act}}$ (*e.g.*, ReLU, Tanh, ELU, and Sigmoid), from which we obtain stereographic activation functions. Stereographic layer normalization $\mathrm{LN}_{\otimes\boldsymbol{\kappa}}$ is defined in the same manner.

**Stereographic logits.** Suppose that $\boldsymbol{x} \in \mathfrak{st}_{\boldsymbol{\kappa}}^d$ is a stereographic embedding retrieved from the last transformer layer. For prediction tasks such as node classification, we need to compute the probability that the node with embedding $\boldsymbol{x}$ belongs to class $c$. Inspired by logistic regression in Euclidean spaces, Bachmann et al. (2020) proposes its stereographic variant as

$$p(y = c \mid \boldsymbol{x}) \propto \exp\left(\mathrm{sign}(\langle -\boldsymbol{p}_c \oplus_{\boldsymbol{\kappa}} \boldsymbol{x}, \boldsymbol{a}_c \rangle)\|\boldsymbol{a}_c\|_{\boldsymbol{p}_c} d_{\boldsymbol{\kappa}}(\boldsymbol{x}, H_{\boldsymbol{a}_c, \boldsymbol{p}_c})\right), \tag{2}$$

where $H_{\boldsymbol{a}_{\boldsymbol{\kappa}}, \boldsymbol{p}_{\boldsymbol{\kappa}}} = \{\boldsymbol{x} \in \mathfrak{st}_{\boldsymbol{\kappa}}^d \mid \langle -\boldsymbol{p}_c \oplus_{\boldsymbol{\kappa}} \boldsymbol{x}, \boldsymbol{a}_c \rangle = 0\}$ is a hyperplane formed by $\boldsymbol{a}_c \in \mathcal{T}_{\boldsymbol{p}_c}\mathfrak{st}_{\boldsymbol{\kappa}}^d$ and $\boldsymbol{p}_c \in \mathfrak{st}_{\boldsymbol{\kappa}}^d$. For stereographic model $\mathfrak{st}_{\kappa}^d$, the distance between $\boldsymbol{x} \in \mathfrak{st}_{\kappa}^d$ and hyperplane $H_{\boldsymbol{a},\boldsymbol{p}}$ equals

$$d_{\kappa}(\boldsymbol{x}, H_{\boldsymbol{a},\boldsymbol{p}}) = \sin_{\kappa*|\kappa|}^{-1}\left(\frac{2|\langle -\boldsymbol{p} \oplus_{\kappa} \boldsymbol{x}, \boldsymbol{a} \rangle|}{(1 + \kappa\|\langle -\boldsymbol{p} \oplus_{\kappa} \boldsymbol{x}, \boldsymbol{a} \rangle\|^2)\|\boldsymbol{a}\|}\right). \tag{3}$$

This distance function can be easily extended to the product-stereographic model as mentioned in Section 3.1, and parameters $\boldsymbol{a}, \boldsymbol{p}$ that define the hyperplane are learnable during training.

## 4.2 Stereographic Multi-Head Attention

Using the stereographic operations above, we propose a multi-head attention mechanism under product-stereographic models. The key intuition is that each $h$-th attention head operates on the $\kappa_h$-stereographic space. Given a sequence of $n$ product-stereographic embeddings $\boldsymbol{X} \in \mathfrak{st}_{\boldsymbol{\kappa}}^{n \times d}$, the attention head with curvature $\kappa$ first obtains values using the stereographic linear layer. For queries and keys, it maps each stereographic embedding to the tangent space of the values as:

$$\boldsymbol{Q} = \boldsymbol{X}\boldsymbol{W}^Q \in \mathcal{T}_{\boldsymbol{V}}\mathfrak{st}_{\kappa}^{n \times d'}, \quad \boldsymbol{K} = \boldsymbol{X}\boldsymbol{W}^K \in \mathcal{T}_{\boldsymbol{V}}\mathfrak{st}_{\kappa}^{n \times d'}, \quad \boldsymbol{V} = \mathrm{Linear}_{\kappa}(\boldsymbol{X};\boldsymbol{W}^V) \in \mathfrak{st}_{\kappa}^{n \times d'}, \quad (4)$$

where $\boldsymbol{W}^Q, \boldsymbol{W}^K \in \mathbb{R}^{d \times d'}$ are the query/key weight matrices, and $\boldsymbol{W}^V \in \mathbb{R}^{d \times d'}$ is the weight matrix for values. Then, the attention score between the $i$-th query $\boldsymbol{Q}_i$ and $j$-th key $\boldsymbol{K}_j$ is computed by parallel-transporting the vectors to the origin, and taking the inner product at the origin as

$$\alpha_{ij} = \langle \mathrm{PT}_{\boldsymbol{V}_i \to \mathbf{0}}(\boldsymbol{Q}_i), \mathrm{PT}_{\boldsymbol{V}_j \to \mathbf{0}}(\boldsymbol{K}_j) \rangle_{\mathbf{0}}. \tag{5}$$

Figure 2 illustrates our geometric attention mechanism. Because the metric tensor of the origin of the stereographic model is simply $4\boldsymbol{I}$ with identity matrix $\boldsymbol{I}$, the Riemannian inner product becomes equivalent to the Euclidean inner product at the origin. Finally, we aggregate values based on the attention scores using the Einstein midpoint:

$$\mathrm{Aggregate}_{\kappa}\left(\boldsymbol{V}, \boldsymbol{\alpha}\right)_i := \frac{1}{2} \otimes_{\kappa} \left(\sum_{j=1}^{n} \frac{\alpha_{ij}\lambda_{\boldsymbol{V}_j}^{\kappa}}{\sum_{k=1}^{n} \alpha_{ik}(\lambda_{\boldsymbol{V}_k}^{\kappa} - 1)} \boldsymbol{V}_j\right), \tag{6}$$

with conformal factor $\lambda_{\boldsymbol{V}_i}^{\kappa}$ at point $\boldsymbol{V}_i \in \mathfrak{st}_{\kappa}^{d'}$. By concatenating the aggregated results from each attention head, the final outcome of product-stereographic multi-head attention is

$$\mathrm{MHA}_{\otimes\boldsymbol{\kappa}}(\boldsymbol{X}) = \|_{h=1}^{H}\mathrm{Aggregate}_{\kappa_h}(\boldsymbol{V}^h, \boldsymbol{\alpha}^h) \in \otimes_{h=1}^{H}\mathfrak{st}_{\kappa_h}^{n \times d}, \tag{7}$$

where $\kappa_h$ denotes the curvature of the $h$-th attention head.

### 4.3 WRAP-UP

For completeness, we fill in the gap on how intermediate steps such as skip-connection are generalized towards non-zero curvatures, and how representations are processed between Transformer layers with distinct curvatures. First, recall that vanilla Transformer utilizes residual connections and Layer normalization to mitigate vanishing gradients and induce better convergence (Vaswani et al., 2017). To apply these operations on representations in the product-stereographic space, we switch to

$$\boldsymbol{X}_l = \mathrm{MHA}_{\otimes\boldsymbol{\kappa}}(\mathrm{LN}_{\otimes\boldsymbol{\kappa}}(\boldsymbol{X}_l^{\mathrm{in}})) \oplus_\kappa \boldsymbol{X}_l^{\mathrm{in}}, \quad \boldsymbol{X}_l^{\mathrm{out}} = \mathrm{FFN}_{\otimes\boldsymbol{\kappa}}(\mathrm{LN}_{\otimes\boldsymbol{\kappa}}(\boldsymbol{X}_l)) \oplus_\kappa \boldsymbol{X}_l. \tag{8}$$

Note that while each attention head in stereographic multi-head attention operates on each stereographic model independently, the product-stereographic feed-forward network $\mathrm{FFN}_{\otimes\boldsymbol{\kappa}}$, for which we use two stereograhpic linear layers with an activation in between, fuses representations from distinct geometries and performs interactions between different steregraphic models similarly to previous work (Zhu et al., 2020b; Deng et al., 2023).

Furthermore, note that each $l$-th Transformer layer operates on a distinct product-stereographic space $\mathfrak{st}^d_{\otimes\boldsymbol{\kappa}^l}$ where $\boldsymbol{\kappa}^l = (\kappa^l_1, \ldots, \kappa^l_H)$ together forms the geometric signature of the layer. For consistency, we assume that the input embeddings are on the product-stereographic model of the first layer (*i.e.*, $\mathfrak{st}^d_{\otimes\boldsymbol{\kappa}^1}$). In case of classification tasks where logits are computed, the product-stereographic logit layer operates on the last set of curvatures (*i.e.*, $\mathfrak{st}^d_{\otimes\boldsymbol{\kappa}^L}$ where $L$ denotes the number of Transformer layers). In between layers, representations are translated from $\mathfrak{st}^d_{\otimes\boldsymbol{\kappa}^l}$ to $\mathfrak{st}^d_{\otimes\boldsymbol{\kappa}^{l+1}}$ by assuming a shared tangent space at the origin (*i.e.*, $\boldsymbol{X}^{\mathrm{in}}_{l+1} = (\exp^{\otimes\boldsymbol{\kappa}^{l+1}}_{\mathbf{0}} \circ \log^{\otimes\boldsymbol{\kappa}^l}_{\mathbf{0}})(\boldsymbol{X}^{\mathrm{out}}_l))$. Altogether, it is straightforward to find that **FPS-T becomes equivalent to the original Transformer as all $\boldsymbol{\kappa}$ approaches 0**, yet it possesses the capability to deviate itself away from Euclidean geometry given it leads to better optimization. For all experiments, we initialize all curvatures as zero to demonstrate the practicality of our method by not requiring additional hyperparameter tuning over different curvature combinations.

### 4.4 EXTENSION TO GRAPH TRANSFORMER

To learn graph-structured data with FPS-T, we borrow the tokenization technique used by TO-KENGT (Kim et al., 2022). Let $\mathcal{G} = (\mathcal{V}, \mathcal{E})$ be a graph with $N$ nodes in node-set $\mathcal{V}$, $M$ edges in edge-set $\mathcal{E}$, and respective features $\boldsymbol{X}^{\mathcal{V}} \in \mathbb{R}^{N \times d}$, $\boldsymbol{X}^{\mathcal{E}} \in \mathbb{R}^{M \times d}$. Then, we tokenize $\mathcal{G}$ into a sequence $\boldsymbol{X} = [\boldsymbol{X}^{\mathcal{V}}, \boldsymbol{X}^{\mathcal{E}}] \in \mathbb{R}^{(N+M) \times d}$ by treating each node and edge as an independent token, and augment the tokens with 1) node identifiers that serve as positional encoding and 2) type identifiers that allow the model to distinguish between node- and edge-tokens. TOKENGT feeds this sequence into vanilla Transformer, an approach proven to pass the 2-dimensional Weisfeiler-Lehman (2-WL) graph isomorphism test and surpass the theoretical expressivity of message-passing GNNs (Kim et al., 2022; Maron et al., 2019). More details on the tokenization procedure can be found in Appendix B.

In our work, we encode the input sequence through FPS-T instead, such that nodes and edges exchange information globally on the product-stereographic space. As augmented feature vectors $\boldsymbol{X}$ are initially Euclidean, we assume each token lies within the tangent space at the origin of the product-stereographic model of the first layer $\mathcal{T}_0\mathfrak{st}^{d'}_{\otimes\boldsymbol{\kappa}^1} \cong \mathbb{R}^{H \times d'}$, where $|\boldsymbol{\kappa}^1| = H$ and $Hd' = d$. Therefore, we apply exponential mapping on the tokens to place them on the product-stereographic model via $\exp^{\otimes\boldsymbol{\kappa}^1}_{\mathbf{0}}(\boldsymbol{X})$, the output of which is forwarded through FPS-T.

### 4.5 COST LINEARIZATION OF STEREOGRAPHIC ATTENTION

One drawback of the graph tokenization method above is that its computational cost becomes intractable when encoding large graphs. As computing the attention score matrix takes time and memory quadratic to the sequence length, a graph with $N$ nodes and $M$ edges incurs an asymptotic cost of $\mathcal{O}((N + M)^2)$, which can be $\mathcal{O}(N^4)$ for dense graphs. Fortunately, there exist various advancements used to make Transformers more efficient (Tay et al., 2022; Kitaev et al., 2020; Choromanski et al., 2020; Wang et al., 2020; Xiong et al., 2021; Cho et al., 2022).

In linearized attention (Katharopoulos et al., 2020), it is shown that the Euclidean attention score $\langle \boldsymbol{Q}_i, \boldsymbol{K}_j \rangle$ can be approximated with the product of kernel function $\phi(\boldsymbol{Q}_i)\phi(\boldsymbol{K}_j)^T$, where $\phi(\boldsymbol{X}) = \mathrm{ELU}(\boldsymbol{X}) + 1$. For stereographic attention (Equation 5), computing dot-products on the tangent space of the origin allows us to extend this kernelization to FPS-T. Let $\tilde{\boldsymbol{Q}}_i = \mathrm{PT}_{\boldsymbol{V}_i \to \mathbf{0}}(\boldsymbol{Q}_i)$ and $\tilde{\boldsymbol{K}}_j = \mathrm{PT}_{\boldsymbol{V}_j \to \mathbf{0}}(\boldsymbol{K}_j)$ be the tangent vectors on the origin prior to taking the dot-product. By applying

Table 1: Synthetic graph reconstruction results in average distortion (lower is better). The best FPS-T configuration and its learned curvatures are well-aligned to the geometry of the input graph.

| Model | Space | TREE | SPHERE | TORUS | RING OF TREES |
|---|---|---|---|---|---|
| TOKENGT | $\mathbb{E}^{10}$ | 0.04363 | 0.04023 | 0.07172 | 0.05553 |
| | $\mathbb{E}^5 \times \mathbb{E}^5$ | 0.04357 | 0.04139 | 0.07167 | 0.05546 |
| FPS-T (ours) | $\mathfrak{st}^{10}_{\kappa_1}$ | **0.00072** | **0.02176** | 0.06415 | 0.03393 |
| | $\mathfrak{st}^5_{\kappa_1} \times \mathfrak{st}^5_{\kappa_2}$ | 0.00105 | 0.02206 | **0.06135** | **0.01630** |
| Best FPS-T curvatures | | $(-1.219)$ | $(+0.0629)$ | $(+1.308, +0.2153)$ | $(+0.3241, -3.314)$ |

| (a) TREE | (b) SPHERE | (c) TORUS | (d) RING OF TREES |
|---|---|---|---|

Figure 3: Illustration of geometric graphs used in our synthetic graph reconstruction experiment.

kernelization to stereographic attention, we can rewrite the stereographic aggregation (Equation 6) as

$$\frac{1}{2} \otimes_\kappa \left( \sum_{j=1}^n \frac{\langle \tilde{Q}_i, \tilde{K}_j \rangle_{\mathbf{0}} \lambda^\kappa_{V_j}}{\sum_{k=1}^n \langle \tilde{Q}_i, \tilde{K}_k \rangle_{\mathbf{0}} (\lambda^\kappa_{V_k} - 1)} V_j \right) \approx \frac{1}{2} \otimes_\kappa \left[ \phi(\tilde{Q}) \left( \phi'(\tilde{K})^T \tilde{V} \right) \right]_i \qquad (9)$$

where $\phi'(K)_i = \phi(K)_i (\lambda^\kappa_{V_i} - 1)$ and $\tilde{V}_i = \frac{\lambda^\kappa_{V_i}}{\lambda^\kappa_{V_i} - 1} V_i$.

This approximation enables FPS-T to encode graphs with $\mathcal{O}(N + M)$ cost, matching the complexity of message-passing GCNs (Wu et al., 2020) while taking the non-Euclidean geometry into account. In Appendix C, we empirically verify this asymptotic cost and also find that the additional cost of Riemannian operations in FPS-T are mostly dominated by pre-existing Transformer operations when encoding large networks. In the upcoming experiments, we use the kernelized approach for FPS-T and find that the approximation performs well in practice.

## 5 EXPERIMENTS

We first evaluate FPS-T on synthetic geometric graph reconstruction (e.g. tree or spherical graph) to verify whether our approach learns curvatures that best fit the input graph. We also benchmark existing graph reconstruction and node classification datasets to empirically demonstrate the benefit of capturing long-range interactions under mixed-curvature spaces in real-world settings.

### 5.1 GRAPH RECONSTRUCTION

**Datasets.** For synthetic graph reconstruction, we generate four types of graphs where the suitable geometry is known a priori — TREES ($\mathbb{H}$), SPHERE ($\mathbb{S}$), TORUS ($\mathbb{S} \times \mathbb{S}$), and RING OF TREES ($\mathbb{S} \times \mathbb{H}$). An example illustration of the synthetic graphs can be found in Figure 3. We then evaluate FPS-T on four real-world networks: WEB-EDU (Gleich et al., 2004) is a web-page network under the *.edu* domain connected with hyperlinks; POWER (Watts & Strogatz, 1998) is a network that models the electrical power grid in western US; BIO-WORM (Cho et al., 2014) is a genetics network of the *C. elegans* worm; FACEBOOK (Leskovec & Mcauley, 2012) is a social network. Further details on the datasets such as sectional curvature statistics of the networks can be found in Appendix D.

**Training.** The goal of graph reconstruction is to learn continuous node representations of the given graph that preserve the edge connectivity structure through distances in the feature space. Let $h_u$ denote the encoded representation of node $u \in \mathcal{V}$ given a graph $\mathcal{G} = (\mathcal{V}, \mathcal{E})$. For synthetic graph reconstruction, we train FPS-T and TOKENGT by minimizing the graph distortion (Gu et al., 2019):

$$\mathcal{L}(h, \mathcal{G}) = \sum_{\substack{(u,v) \in \mathcal{V} \times \mathcal{V} \\ u \neq v}} \left| \left( \frac{d(h_u, h_v)}{d_\mathcal{G}(u, v)} \right)^2 - 1 \right|$$

where $d(h_u, h_v)$ denote the distance between $h_u$ and $h_v$ on the representation space, and $d_\mathcal{G}(u, v)$ equals the shortest path distance between nodes $u$ and $v$ on graph $\mathcal{G}$. Both methods use a single layer with 1 or 2 attention heads with a combined latent dimension of 10.

| Dataset | WEB-EDU | POWER | FACEBOOK | BIO-WORM |
|---|---|---|---|---|
| Avg. Curvature | -0.63 | -0.28 | -0.08 | -0.03 |
| MLP | $83.24_{\pm1.32}$ | $83.89_{\pm4.02}$ | $50.64_{\pm15.12}$ | $73.34_{\pm20.85}$ |
| GCN | $79.95_{\pm0.23}$ | $98.25_{\pm0.02}$ | $78.99_{\pm0.29}$ | $93.32_{\pm1.06}$ |
| GAT | $88.86_{\pm0.36}$ | $99.03_{\pm0.01}$ | $82.81_{\pm0.25}$ | $97.76_{\pm0.03}$ |
| SAGE | $86.34_{\pm0.31}$ | $97.58_{\pm0.14}$ | $81.01_{\pm0.26}$ | $96.86_{\pm0.06}$ |
| SGC | $78.78_{\pm0.12}$ | $97.69_{\pm0.05}$ | $74.69_{\pm0.36}$ | $89.73_{\pm0.59}$ |
| TOKENGT | $89.45_{\pm0.06}$ | $99.10_{\pm0.00}$ | $84.71_{\pm0.02}$ | $97.82_{\pm0.02}$ |
| HGCN | $80.13_{\pm0.31}$ | $96.82_{\pm0.08}$ | $74.35_{\pm5.39}$ | $86.96_{\pm0.30}$ |
| HGNN | $83.64_{\pm0.26}$ | $97.85_{\pm0.05}$ | $78.74_{\pm0.58}$ | $90.97_{\pm1.06}$ |
| HAT | $90.21_{\pm0.36}$ | $93.86_{\pm0.34}$ | $80.09_{\pm0.20}$ | $93.58_{\pm0.42}$ |
| $\kappa$-GCN | $55.34_{\pm35.88}$ | $98.23_{\pm0.09}$ | $20.80_{\pm20.69}$ | $84.16_{\pm13.67}$ |
| $\mathcal{Q}$-GCN | $80.34_{\pm0.07}$ | $97.87_{\pm0.01}$ | $76.33_{\pm0.01}$ | $96.15_{\pm0.01}$ |
| FPS-T | $\mathbf{99.10_{\pm0.01}}$ | $\mathbf{99.32_{\pm0.01}}$ | $\mathbf{86.16_{\pm0.10}}$ | $\mathbf{98.19_{\pm0.03}}$ |

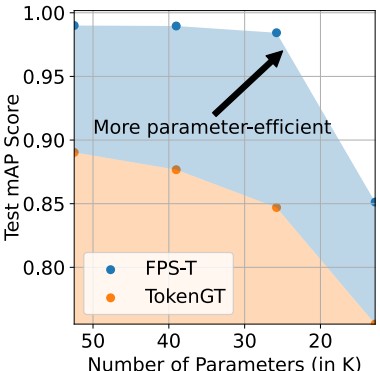

Figure 4: **Left:** Real-world graph reconstruction results. We run each method under 5 random seeds and report the average mAP with 95% confidence intervals. **Right:** Test mAP (Y-axis) of FPS-T and TOKENGT on WEB-EDU with decreasing model size (X-axis; by decreasing the latent dimension). Using mixed-curvarture spaces can be more parameter efficient in preserving graph structures.

For real-world graph reconstruction, we instead minimize a loss function that aims for preserving the local connections as computing the all-pairwise shortest path distances becomes computationally intractable with large networks:

$$\mathcal{L}(\boldsymbol{h}, \mathcal{G}) = \sum_{(u,v)\in\mathcal{E}} \log \frac{e^{-d(\boldsymbol{h}_u, \boldsymbol{h}_v)}}{\sum_{v'\in\bar{\mathcal{E}}(u)} e^{-d(\boldsymbol{h}_u, \boldsymbol{h}_{v'})}}$$

Here, $\bar{\mathcal{E}}(u)$ denotes the set of non-neighbors of node $u$. In addition to TOKENGT, we also compare FPS-T against baselines including Euclidean (GCN (Kipf & Welling, 2016), GAT (Veličković et al., 2017), SAGE (Hamilton et al., 2017), SGC (Wu et al., 2019)), hyperbolic (HGCN (Chami et al., 2019), HGNN (Liu et al., 2019), HAT (Zhang et al., 2021)), and mixed-curvature ($\kappa$-GCN (Bachmann et al., 2020), $\mathcal{Q}$-GCN (Xiong et al., 2022)) message passing-based GCNs. For fair comparison, we set the number of layers to one and latent dimension to 16 for all models. We train all models for 10k epochs using an Adam optimizer with learning rate $1e-2$. The node features are given as one-hot encodings with additional random noise following Xiong et al. (2022). We defer details on the choice of hyperparameters of baseline methods to Appendix E.

**Results.** Table 1 reports the synthetic graph reconstruction results in average graph distortion as well as curvatures learned by FPS-T. As expected, FPS-T consistently outperforms its Euclidean counterpart on all four networks due to the networks exhibiting highly non-Euclidean structures. Despite being initialized at zero, the learnable curvatures in FPS-T converge towards curvatures that intuitively match with the input graph: for RING OF TREES, FPS-T with two attention heads converge towards one positive and one negative curvature, outperforming the single-head variant.

Next, the left table in Figure 4 shows the average sectional curvature of each real-world network and corresponding graph reconstruction results in mean average-precision (mAP) which measures the average ratio of nearest points that are actual neighbors of each node. We find that FPS-T shows significant performance gains on all four networks when compared to all baselines including Euclidean TOKENGT. Specifically, FPS-T shows a 10.5% gain in mAP against TOKENGT on WEB-EDU with an average sectional curvature of -0.63, showing that performing attention on the non-Euclidean product-stereographic space is especially effective when encoding graphs containing of many non-zero sectional curvatures.

Note that non-Euclidean spaces are theoretically known to well-embed complex structures in low dimensions, while Euclidean spaces require a large number of dimensions to attain reasonable precision (Sala et al., 2018). Based on this observation, we test whether FPS-T enjoys better parameter efficiency compared to TOKENGT by training two models with decreasing latent dimensions in $\{16, 12, 8, 4\}$. In the right plot of Figure 4, we report the mAP score of TOKENGT and FPS-T on the WEB-EDU network after training with decreasing number of parameters. We observe that our approach of incorporating mixed-curvature spaces consistently obtains low distortion embeddings in a more parameter-efficient manner, outperforming TOKENGT with $d = 16$ using half its model size.

Table 2: Node classification results. We run each method under 10 different random seeds and report the average F1 scores with 95% confidence intervals and average rankings across all datasets.

| Dataset $\mathcal{H}(\mathcal{G})$ | TEXAS 0.11 | CORNELL 0.13 | WISCONSIN 0.20 | ACTOR 0.22 | AIRPORT 0.72 | CITESEER 0.74 | PUBMED 0.80 | CORA 0.81 | Avg. Rank |
|---|---|---|---|---|---|---|---|---|---|
| MLP | $70.54_{\pm3.00}$ | $58.38_{\pm4.04}$ | $81.20_{\pm1.87}$ | $33.62_{\pm0.55}$ | $54.05_{\pm1.78}$ | $52.58_{\pm1.97}$ | $67.17_{\pm0.91}$ | $52.44_{\pm1.08}$ | 8.25 |
| GCN | $57.84_{\pm1.62}$ | $47.84_{\pm1.77}$ | $45.40_{\pm2.62}$ | $27.09_{\pm0.36}$ | $92.00_{\pm0.63}$ | $71.38_{\pm0.43}$ | $78.37_{\pm0.26}$ | $80.40_{\pm0.53}$ | 7.38 |
| GAT | $59.46_{\pm1.12}$ | $55.14_{\pm1.80}$ | $46.20_{\pm2.30}$ | $27.43_{\pm0.23}$ | $92.35_{\pm0.36}$ | $71.70_{\pm0.28}$ | $78.14_{\pm0.31}$ | $82.29_{\pm0.46}$ | 6.13 |
| SAGE | $68.38_{\pm3.54}$ | $70.54_{\pm2.01}$ | $78.40_{\pm0.52}$ | $36.87_{\pm0.50}$ | $93.21_{\pm0.57}$ | $70.58_{\pm0.42}$ | $77.31_{\pm0.59}$ | $78.88_{\pm0.87}$ | 5.13 |
| SGC | $57.57_{\pm2.96}$ | $52.97_{\pm2.87}$ | $46.40_{\pm2.01}$ | $27.14_{\pm0.46}$ | $90.48_{\pm1.01}$ | $\mathbf{72.11_{\pm0.38}}$ | $75.11_{\pm1.27}$ | $79.68_{\pm0.65}$ | 8.25 |
| TOKENGT | $88.65_{\pm2.06}$ | $71.62_{\pm2.13}$ | $83.00_{\pm0.65}$ | $36.59_{\pm0.89}$ | $95.90_{\pm0.59}$ | $71.23_{\pm0.51}$ | $\mathbf{78.93_{\pm0.27}}$ | $81.42_{\pm0.79}$ | 2.50 |
| HGCN | $54.59_{\pm3.93}$ | $55.68_{\pm1.80}$ | $55.60_{\pm2.53}$ | $28.89_{\pm0.16}$ | $92.47_{\pm0.63}$ | $69.92_{\pm0.61}$ | $75.67_{\pm0.99}$ | $80.00_{\pm0.85}$ | 7.00 |
| HGNN | $50.81_{\pm3.60}$ | $52.70_{\pm1.42}$ | $54.60_{\pm2.68}$ | $29.09_{\pm0.19}$ | $90.55_{\pm0.71}$ | $69.82_{\pm0.53}$ | $76.72_{\pm0.86}$ | $79.30_{\pm0.51}$ | 8.75 |
| HAT | $82.16_{\pm2.52}$ | $70.54_{\pm1.67}$ | $81.80_{\pm1.36}$ | $38.34_{\pm0.26}$ | $92.88_{\pm0.57}$ | $68.14_{\pm0.53}$ | $77.50_{\pm0.42}$ | $79.81_{\pm0.58}$ | 4.38 |
| $\kappa$-GCN | $56.22_{\pm4.38}$ | $55.68_{\pm5.59}$ | $46.60_{\pm2.41}$ | $26.39_{\pm0.60}$ | $82.58_{\pm3.70}$ | $54.06_{\pm4.45}$ | $68.61_{\pm3.05}$ | $73.70_{\pm0.69}$ | 10.3 |
| $\mathcal{Q}$-GCN | $51.35_{\pm3.44}$ | $55.95_{\pm2.85}$ | $52.80_{\pm2.20}$ | $28.18_{\pm0.55}$ | $91.39_{\pm1.05}$ | $66.15_{\pm0.45}$ | $77.13_{\pm0.59}$ | $79.63_{\pm0.57}$ | 8.25 |
| FPS-T | $\mathbf{89.19_{\pm2.37}}$ | $\mathbf{72.16_{\pm2.96}}$ | $\mathbf{83.60_{\pm1.14}}$ | $\mathbf{39.61_{\pm0.54}}$ | $\mathbf{96.01_{\pm0.55}}$ | $70.03_{\pm0.71}$ | $78.52_{\pm0.58}$ | $\mathbf{82.32_{\pm0.70}}$ | 1.75 |

## 5.2 NODE CLASSIFICATION

**Datasets.** For node classification we experiment on eight different networks: three WebKB networks (TEXAS, CORNELL, WISCONSIN) that connect web-pages via hyperlinks (Craven et al., 1998), a co-occurrence network from Wikipedia pages related to English films (ACTOR) (Tang et al., 2009), three citation networks (CITESEER, PUBMED, CORA) (Sen et al., 2008), and an airline network (AIRPORT) (Chami et al., 2019). These networks are chosen to test our approach under a wide spectrum of graph homophily $\mathcal{H}(\mathcal{G})$, which measures the ratio of edges that connect nodes that share the same label (Zhu et al., 2020a). In other words, a hetereophilic graph with small graph homophily requires capturing long-range interactions for proper labeling, which is naturally difficult for message passing-based approaches with small receptive fields. More detailed statistics on the networks can be found in Appendix D.

**Training.** For all methods, we fix the embedding dimension to 16 and train each model to minimize the cross-entropy loss using an Adam optimizer with a learning rate of $1e{-}2$. For models that use learnable curvatures (*i.e.*, HGCN, $\kappa$-GCN and FPS-T), we use a learning rate of $1e{-}4$ for the curvatures. The optimal number of layers, activation function, dropout rate, and weight decay of each method are chosen via grid search on each dataset. Details on the hyperparameter search-space and dataset splits can be found in Appendix E.2.

**Results.** Table 2 shows the results from node classification. Overall, our method attains best accuracy on 6 out of 8 datasets, showing that FPS-T is effective across networks with various graph homophily. In case of hetereophilic networks, we find that the small receptive fields of message-passing GCNs are extremely inadequate, often being outperformed by a simple MLP that completely ignores the graph connectivity. On the other hand, FPS-T consistently outperforms MLP as well as GCN baselines, due to its ability to exchange information across long distances via global-attention. It also significantly outperforms TOKENGT by 8.3% on Actor, showing that adjusting the geometry towards non-Euclidean can further enhance predictive performance. In homophilic networks where message-passing is more well-suited, FPS-T shows competitive performance against GCN baselines. This is expected as FPS-T enjoys the same capacity as TOKENGT to mimic any order-2 equivariant bases (Kim et al., 2022), which includes local message-passing, through attention score computation.

## 6 CONCLUSION

We propose FPS-T, a natural generalization of the Transformer architecture towards mixed-curvature spaces with learnable curvatures. When combined with the graph tokenization technique of Kim et al. (2022), our model can embed graphs with less distortion and higher parameter-efficiency than its Euclidean counterpart by operating on the product-stereographic model. We also show that our model outperforms existing hyperbolic and mixed-curvature message-passing GCN baselines on node classification via global-attention that can capture long-range interactions. By linearizing the cost of self-attention through kernelized approximation, FPS-T runs in cost linear to the number of nodes and edges, allowing practical use on large-scale networks. For future work, we plan to extend towards heterogeneous manifolds (Giovanni et al., 2022) with input-dependent sectional curvatures and also optimize Riemannian operations towards better stability and efficiency under finite precision. As we propose a foundational generalization of the Transformer architecture, investigating what geometry suits best for various tasks in the NLP and CV domain would also be an interesting direction.

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
