# OpenReview forum: "Curve Your Attention: Mixed-Curvature Transformers for Graph Representation Learning"
_ICLR.cc/2024/Conference — ICLR 2024 Conference Withdrawn Submission_

### Official Review · Reviewer_g83M · 2023-10-31

**Soundness:** 2 fair
**Presentation:** 2 fair
**Contribution:** 1 poor
**Rating:** 1
**Confidence:** 5

**Summary:**

The authors propose a Product-Stereographic Transformer, a generalization of Transformers towards operating on the product of constant curvature spaces. The work also provides a kernelized approach to non-Euclidean attention for further efficiency. The authors perform various experiments on graph reconstruction and node classification to demonstrate their model performance.

**Strengths:**

1. The paper is overall well-organized and the writing is easy to follow.

2. The authors evaluate on many datasets and models to showcase their model performance.

**Weaknesses:**

1. In the Related Work section, the authors are missing reference to a fundamental recent paper learning jointly on both hyperbolic and spherical spaces in a unified end-to-end pipeline:
KDD 2022: Iyer et al. 2022. Dual-Geometric Space Embedding Model for Two-View Knowledge Graphs. In Proceedings of the 28th ACM SIGKDD Conference on Knowledge Discovery and Data Mining (KDD '22). Association for Computing Machinery, New York, NY, USA, 676–686. https://doi.org/10.1145/3534678.3539350

2. In the Preliminaries, the authors should also provide formulas for the retraction operators including interpretation of the exponential mapping and log mapping operations e.g., why the Euclidean tangent space is needed etc.

3. In the Experiments section, it seems strange that the authors are utilizing synthetic geometric graph datasets, when their motivation was that hierarchies and cycles are commonly found in real-world datasets. As such, the motivation behind using Table 1 needs to be defined better.

4. The novelty of this work is also lacking. Chami et. al, already propose the using of retraction operators (exponential and log mapping that can be generalized to any non-Euclidean geometric space of constant curvature K. Further, product spaces have also already been proposed. Moreover, even integrating both hyperbolic and spherical spaces jointly has been proposed by Iyer et. al. This work just seems to be a combination of all of the above works.

**Questions:**

The design choice behind why the authors are using a product-stereographic space to model various topologies of the data is also unclear. Why not consider embeddings different topologies in different spaces instead of just considering one product space? It seems to me that this direction has not even been explored.

---

### Official Review · Reviewer_c5LD · 2023-10-31

**Soundness:** 3 good
**Presentation:** 3 good
**Contribution:** 3 good
**Rating:** 5
**Confidence:** 4

**Summary:**

This paper proposes a  for generalizing Transformer architectures to operate on non-Euclidean spaces. The main contributions are:

- Proposes FPS-T, which generalizes Transformers to operate on the product-stereographic model. This allows each layer to learn curvature values for different attention heads.

- Applies FPS-T to graph representation learning by integrating it with the Tokenized Graph Transformer. It uses a kernelized approximation to reduce computational complexity.

- Evaluates FPS-T on synthetic and real-world graph reconstruction and node classification tasks. Finds it can learn suitable curvatures and outperform Euclidean Transformers.

**Strengths:**

The research proposes an exciting and novel extension of Transformers to non-Euclidean geometry, which has not been explored before. Generalizing attention mechanisms to curved spaces is a contribution.

**Weaknesses:**

- The motivation for non-Euclidean Transformers is not fully justified. The introduction claims they are necessary for modeling hierarchical and cyclic graphs but does not provide compelling evidence current Euclidean Transformers fail on such structures. More analysis on the limitations of existing methods is needed.

1. The stereographic model limits flexibility compared to input-dependent curvatures. Methods like heterogeneous manifolds (cited in the paper) can adapt curvature per node/edge based on features. Fixing curvature by attention head may be too restrictive. The paper could experiment with more adaptive curvature mechanisms.

2. Scalability is a concern. The largest graph evaluated has only 4,000 nodes and 88k edges. More experimentation on large real-world networks is important to demonstrate practical value.

3. No transformer-based baselines are compared.

4. Ablation studies could provide more insight. For example, how do learned curvatures evolve during training? How sensitive is performance to curvature initialization? Are some attention heads more "non-Euclidean" than others?

**Questions:**

You claim Euclidean Transformers are inadequate for modeling hierarchical and cyclic graphs. However, recent works like Graphormer show strong performance on tasks like molecular property prediction that involve such structures. Can you provide more concrete evidence on the limitations of existing methods? Comparisons to recent graph Transformers on suitable benchmarks would help make this case.


Have you experimented with more adaptive, input-dependent curvature mechanisms? Fixing curvature by attention head seems restrictive. Heterogeneous manifolds allow varying curvature per node/edge. How do learned curvatures in FPS-T compare to feature-based curvature?

---

### Official Review · Reviewer_Wzei · 2023-11-01

**Soundness:** 2 fair
**Presentation:** 3 good
**Contribution:** 2 fair
**Rating:** 5
**Confidence:** 4

**Summary:**

The authors present a new global attention-based graph Transformers that operates entirely on the product of spaces with constant curvature, relying on stereographic product models. Building on these global attention mechanisms, the authors extend tokenized graph transformers from a Euclidean framework (TokenGT) to a non-Euclidean framework (FPS-T). They further consider the approximation of pairwise products in attention layers using a popular linear approximation technique that mimics feature maps, so that FPS-T becomes linear in the number of nodes and edges instead of quadratic. They then compare these two transformers on graph reconstruction tasks considering synthetic and real datasets. Finally, they evaluate FPS-T on 8 well-known node classification tasks, including homophilic and heterophilic graphs. FPS-T outperforms the compared methods on both task types.

**Strengths:**

-	Overall the paper is well-written.
-	The proposed transformer is new and relevant
-	Synthetic graph reconstruction experiments on specific geometry are relevant and show that FPS-T outperforms its Euclidean counterpart Token-GT.
-	FPS-T is shown to outperform several methods on the reconstruction of real networks, especially when graphs have significantly negative sectional curvatures e.g lines, cycles and trees.
-	FPS-T outperforms benchmarked methods on 6 out of 8 node classification tasks, especially in heterophilic settings.

**Weaknesses:**

-	1. **Rather incremental** The design of FPS-T is clearly an adaptation of the work of (Bachmann & al, 2020) to transformer-like architectures.
-	2. **No theoretical insights** There are no significant theoretical analysis that would provide insights on specific features of FPS-T: e.g when graphs whose sectional curvature distribution has a large variance what would be a good constant curvature space (or product of spaces) to discriminate them ?
-	3. **Few unclear parts remain**:
	 - i) I think that equation 4 is wrong: expressions of Q and K seem wrong as a such linear transformation will not embed a stereographic embedding in the tangent space in V. are there missing log_V mappings ?
         - ii) Equation 6 is not clearly explained. Moreover a conformal factor of 1 seems to lead to a ill-defined aggregation function, this point should be clarified by authors.
         - iii) Could you further characterize the resulting function $exp_0( f_{act} (log_0))$ when $f_{act}$ is not differentiable everywhere as the ReLU activation function ?
         - iv) Section 4.5 is not totally clear: I believe that the ‘kernelization’ aims at approximating the softmax activation $\sigma(< Q_i, K_j>)$ not just the inner product $< Q_i, K_j>$ otherwise I do not see the point. Moreover no context is provided w.r.t the current SOTA to reduce the computational cost of transformers.
-	4. **Incomplete experiments and analysis**:
        - i) Under exploited synthetic experiments: could you provide curvatures for the designed graphs so that we can quantify their correspondences to estimated constant curvatures in FPS-T? Could you perform a sensitivity analysis w.r.t the embedding dimensions and the number of considered heads for both FPS-T and Token_GT?
         - ii) No ablation study w.r.t learning the curvatures instead of taking fixed ones, e.g according to modes of the estimated sectional curvatures. This could be at least considered on the toy datasets.
         - iii) Potential fairness issues in the benchmarks on real-world networks which should be discussed by authors: a) Most of benchmarked methods depend on a considerably fewer hyperparameters and the validation grid seems to considerably differ from original paper, while also fitting transformer-based approaches better. Moreover these methods tend to be considerably faster than FPS-T so fitting original validation grids in the paper seems affordable. Could authors provide performances distributions across validated hyperparameters to get an idea of the method robustness? ; b) Laplacian Eigenvectors are by default included in transforms and no considered for GNNs while there are Spectral augmentation techniques that could also be used.; c) the duality w.r.t hops vs number of layers is clearly different between GNN-based and transformer-based approaches. I would suggest to use e.g Jumping Knowledge concatenation of embeddings for both methods to relax the dependency to a well-chosen validated hyperparameter.
        - iv ) SOTA methods for node classification tasks are not present in the benchmark and clearly seem to outperform FPS-T e.g [A, B]


[A] Luan, S., Hua, C., Lu, Q., Zhu, J., Zhao, M., Zhang, S., ... & Precup, D. (2022). Revisiting heterophily for graph neural networks. Advances in neural information processing systems, 35, 1362-1375.

[B] He, M., Wei, Z., & Xu, H. (2021). Bernnet: Learning arbitrary graph spectral filters via bernstein approximation. Advances in Neural Information Processing Systems, 34, 14239-14251.

**Questions:**

I invite the authors to discuss the above-mentioned weaknesses and to answer the questions (potentially implying additional experiments) I have associated with them in order to complete my development.

---

### Author Response · Authors · 2023-11-23
**Common Response to All Reviewers**

We sincerely thank the reviewers for your constructive feedback. After careful consideration, we have decided to withdraw our submission, but we wish to use this opportunity to clarify our current contribution as well as summarize key improvements suggested by the reviewers towards advancing our work.

To sum up, our FPS-T framework is the first mixed-curvature graph Transformer model that displays two major features:

- To benefit from using mixed-curvature geometry to the fullest extent, FPS-T operates entirely on the product-stereographic space, and each component of FPS-T is carefully designed to be mathematically sound and numerically stable, both of which are crucial for proper learning.
- All curvatures in FPS-T are learnable without extensive hyperparamter tuning, embracing vanilla Transformer as a special case when all curvatures are fixed as zero. This implies that FPS-T inherits the same high theoretical expressiveness previously demonstrated with Transformers [A, B] and allows practitioners to adopt techniques from efficient Transformer literature to reduce the computational cost [C, D, E].

Our technical contributions relevant to the features above are three-fold, each of which are yet unexplored in previous work:

- **Stereographic multi-head attention (Section 4.2)**. Our stereographic attention module views each query/key vector as a tangent vector within the tangent bundle of the manifold. The attention-scores are computed by parallel-transporting all query-key vectors to the origin, and then taking the inner-product via the metric tensor at the origin (Figure 2). Our work is the first to propose a geometric interpretation of widely used dot-product attention, and to generalize it towards mixed-curvature spaces for geometry-aware feature aggregation.
- **Generalization of other components (Sections 4.1 and 4.3)**. To induce full operation of FPS-T on the mixed-curvature space, further generalization steps include consistently decomposing the feature dimension into equal parts within the feed-forward block and assuming a common tangent space at the origin between two Transformer layers with different curvature signatures. Each individual operation is carefully generalized such that it is geometrically coherent yet inclusive of its corresponding operation in vanilla Transformer.
- **Cost linearization under mixed-curvature geometry (Section 4.5)**. As FPS-T generalizes Transformers, it permits seamless application of efficient Transformer methods to reduce the cost of self-attention. For this work, we find that the overall cost can be linearized via kernelization [E] while respecting the non-Euclidean geometry by properly scaling the key/value vectors based on curvature-dependent conformal factors (Eqn. 11). With this modification, the cost of FPS-T becomes linear to the number of nodes and edges (same as message-passing GNNs) which grants applicability to large-scale networks.

Based on the reviewers' constructive feedback, we plan to improve our work by providing more theoretical insights for clearer motivation [Wzei, c5LD], performing ablation studies to analyze the learnable curvatures with respect to sectional curvatures present in the input graph [Wzei, c5LD], and migrating towards heterogeneous manifolds [c5LD] or different topological spaces [g83M] for better flexibility. We will also add suggested references, baselines, and experiments on large-scale or molecular property prediction benchmarks [Wzei, c5LD, g83M].

Thank you again for your time and commitment.

Sincerely, Authors of Submission5253.